# Graphene for Antimicrobial and Coating Application

**DOI:** 10.3390/ijms23010499

**Published:** 2022-01-02

**Authors:** Viritpon Srimaneepong, Hans Erling Skallevold, Zohaib Khurshid, Muhammad Sohail Zafar, Dinesh Rokaya, Janak Sapkota

**Affiliations:** 1Department of Prosthodontics, Faculty of Dentistry, Chulalongkorn University, Bangkok 10330, Thailand; viritpon.s@chula.ac.th; 2Department of Oral Biology, Faculty of Dentistry, University of Oslo, Sognsvannsveien 10, 0316 Oslo, Norway; herlings7b@msn.com; 3Department of Prosthodontics and Implantology, College of Dentistry, King Faisal University, Al-Hofuf 31982, Al Ahsa, Saudi Arabia; zsultan@kfu.edu.sa; 4Department of Restorative Dentistry, College of Dentistry, Taibah University, Al Madinah 41311, Al Munawwarah, Saudi Arabia; drsohail_78@hotmail.com; 5Department of Dental Materials, Islamic International Dental College, Riphah International University, Islamabad 44000, Pakistan; 6Department of Clinical Dentistry, Walailak University International College of Dentistry, Walailak University, Bangkok 10400, Thailand; 7Research Center, UPM Pulp Research and Innovations, 53200 Lappeenranta, Finland

**Keywords:** graphene, coatings, bioactivity, tissue engineering, bone regeneration

## Abstract

Graphene is a versatile compound with several outstanding properties, providing a combination of impressive surface area, high strength, thermal and electrical properties, with a wide array of functionalization possibilities. This review aims to present an introduction of graphene and presents a comprehensive up-to-date review of graphene as an antimicrobial and coating application in medicine and dentistry. Available articles on graphene for biomedical applications were reviewed from January 1957 to August 2020) using MEDLINE/PubMed, Web of Science, and ScienceDirect. The selected articles were included in this study. Extensive research on graphene in several fields exists. However, the available literature on graphene-based coatings in dentistry and medical implant technology is limited. Graphene exhibits high biocompatibility, corrosion prevention, antimicrobial properties to prevent the colonization of bacteria. Graphene coatings enhance adhesion of cells, osteogenic differentiation, and promote antibacterial activity to parts of titanium unaffected by the thermal treatment. Furthermore, the graphene layer can improve the surface properties of implants which can be used for biomedical applications. Hence, graphene and its derivatives may hold the key for the next revolution in dental and medical technology.

## 1. Introduction

Graphene, having a sp^2^ configuration, is made from a thin sheet of carbon atoms (Figure 1) [1,2,3]. The various forms of graphene include pure/pristine graphene, graphene oxide (GO) containing –COC–, –COOH, or –COH, reduced GO (rGO), and animated graphene oxide (AGO). Graphene materials have outstanding various properties with good mechanical strength, high surface area, elasticity, stiffness, excellent biocompatibility, superior electrical and thermal conductivity, and ease of functionalization [4,5,6,7,8]. Therefore, graphene is attractive in different fields including medicine and dentistry [3,9,10,11]. The review aims to present an introduction of graphene and presents a comprehensive up-to-date review of graphene as an antimicrobial and coating application in medicine and dentistry.

## 2. Production of Graphene

Different grades of graphene can be prepared by various methods of production depending on the type of application. Such methods of production include mechanical exfoliation of graphite, epitaxial growth of graphene, liquid-phase exfoliation (LPE), chemical vapor deposition, and molecular assembly [3,4,13]. The most common methods are shown in Figure 2.

The simplest method of production of graphene is by mechanical exfoliation in which the graphite is subjected to tape exfoliation followed by transfer of graphene to a substrate [14,15]. Through this method, the greatest quality of graphene is produced, however, to scale-up the process is not possible [16]. The characteristics of graphene produced from various methods differ, as shown in Table 1. Graphene can grow epitaxially on SiC (silicon carbide) at high-temperature (1300–1800 °C) [17]. This method produces atomically smooth graphene nanostructures but may contain certain manufacturing defects. Furthermore, molecular assembly induces modulation of graphene using metal phthalocyanines [18] which is effective to improve the electronic properties [19,20] and the molecular ordering is critical to achieving potential shapes [19]. Liquid phase extraction (LPE) is important for the mass manufacture of graphene [21,22]. Common reported techniques of LPE include sonication [23], jet cavitation [24], micro-fluidization [25], and high-shear mixing [22]. Sonication can produce high concentrations of monolayer to few-layer graphene [23,26]. The factors responsible for the graphene exfoliation include the sonication process, shear forces, the dispersion medium, and the centrifugation process [27,28,29]. Graphene is also grown on non-metallic substrates such as SiO_2_, h-BN, or quartz, using chemical vapor deposition, which allows direct deposition of high-quality graphene [30,31]. Chemical vapor deposition of graphene can result in 3D structures having low density, high surface area, and fast electron transport [32,33,34]. These properties are suitable for engineering, nanotechnology, and biomedical applications.

## 3. Structure and Properties of Graphene

A flat, 2D, sheet of graphene is single to multi-layered while graphene 3D structures can be produced to take various forms (flakes, foams, shells, and hierarchical structures) [32,35]. A graphene film may be comprised of a monolayer, bilayer, or multi-layer. Monolayer graphene is very thin (0.35 ± 0.01 nm) [36] and multilayer graphene has <10 layers [6], as reported by Raman scattering, scanning probe microscopy, and optical contrast [37]. The 2D graphene layers can have a pore size of less than a millimeter which can subsequently be incorporated into porous 3D graphene forms [38]. The 3D foam structures have a larger surface area, strength, are stiff, lightweight, and provide excellent electronic and thermal conductivity, and pathways for ionic transport.

The structure of GO and rGO and their process of production is shown in Figure 3 [39]. Generally, GO is manufactured by the oxidation of graphite from Hummers’ method [40,41]. By thermal-, chemical-, and electrochemical reduction, GO yields rGO. GO and rGO have functional capabilities and wider applications beyond that of pristine graphene [3,42,43].

The AGO can be produced from the reduction and amination of graphene oxide via two-step liquid phase treatment with hydrobromic acid and ammonia solution in mild conditions [44]. The AGO is biocompatible, has electrical conductivity, and has the tendency to form wrinkled and corrugated graphene layers are observed in the AGO derivative compared to the pristine rGO. AGO can be used for biosensing, photovoltaic, catalysis application, and is used as a starting material for further chemical modifications.

Table 2 shows the essential properties of graphene, GO, and rGO [3,43]. Graphene has good electron mobility [45], increased surface area [46], good electrical conductivity [15], good thermal conductivity [47], high elastic modulus [48], strength and stiffness [48], and good wear and friction properties [4,7]. Large surface area and the ability to form nanocomposites, graphene-based materials have wide applications in regenerative medicine and drug delivery. High strength, wear-resistant, and low friction are useful in coatings and nanocomposites. Good electrical property is suitable for biosensors, semiconductors, and supercapacitors.

## 4. Characterization and Properties of Graphene

Several methods to study graphene’s surface structure are of use, these include transmission electron microscopy, scanning electron microscope, energy dispersive spectroscopy, Raman spectroscopy, X-ray diffraction, and atomic force microscopy, [49,50,51,52,53]. A notable characteristic of GO includes a somewhat rough surface morphology, as observed using a scanning electron microscope [52,53]. Raman spectra of graphene-based materials exhibit a D- and G band at about 1320/cm and 1570/cm, respectively [54]. D bands signify the breathing mode of κ-point phonons with A*_1g_* symmetry and the G band signifies the tangential stretching mode of the E*_2g_* phonon of the carbon sp^2^ atoms. The I_D_/I_G_ ratio of around 0.84–0.97 has been reported [49].

The crystallinity and spacing of the interplane of graphene can be studied from XRD. The deflection, height, and 3D images are obtained at micron and nanoscale, by the XRD. AFM can reveal the surface structure and allow for the observation of features at the molecular and atomic levels. R_a_ (roughness average) can be also calculated from the AFM. XPS makes it possible to study the binding between C−O−C and C−C, and elemental composition [55].

## 5. Functionalization of Graphene

The development of nanocomposites has a long history. Although graphene has potential applications in engineering and biomedicine, its properties can be further improved via functionalization and doping due to its sp2 carbon atoms [56,57,58]. Graphene-based materials can be strengthened by various biopolymers (e.g., epoxy and polyketone) and metals (e.g., Zr, Ag, Cu, Zn, Au, Al, Ni, and Mg) [9,59,60,61,62], and nonmetals. As graphene is atomically thin, flat, and conducting material, it is suitable to produce energy storage devices [63]. At, present, the biomedical application of graphene nanocomposites is increasing [9,61,64,65]. The graphene nanocomposites have improved biocompatibility [66,67,68], surface properties [3,60], and mechanical properties [60] compared to pristine graphene. Graphene oxide, which is more amenable to chemical modification than pristine graphene. These properties permit applications involving protective and anticorrosion coatings [67,68], friction reduction [60], and antibacterial utilizations [69].

In graphene, n- or p-type doping Fermi level production is generally seen by physical or chemical bonds [70,71]. In graphene, chemical functionalization offers an obvious solution to the problems associated with graphene [72]. Electron-donating or -withdrawing groups can be bonded to the graphene network by synthetic chemistry methods, which could contribute to the bandgap widening and good dispersibility in common organic solvents.

The functionalization of graphene can be through covalent or non-covalent. Covalent bonds with graphene can occur using radical species, including nitrene, carbene, and aryl intermediates [72]. Conversely, modification of graphene occurs through noncovalent interactions, such as π–π interactions, van der Waals forces, ionic interactions, and hydrogen bonding, and result in major alteration of its structure and electronic properties [72]. The noncovalent interaction of graphene occurs with aromatic species, organic molecules, other carbon nanostructures, and inorganic species.

An aryl group can be grafted on the sp^2^ carbon network of graphene using a diazonium salt and this has been widely applied to form covalently functionalized conducting or semiconducting materials [73,74]. A dinitrogen molecule is eliminated, and then, an electron is transferred from graphene to the diazonium salt to form an aryl radical. Thionine (Th) diazonium cation—covalently attached to the glassy carbon (GC) electrode via graphene nanosheets (GNs) (GC–GNs–Th)—has potential for application in sensors for detecting glucose and nitrite [74]. In addition, perfluorophenyl azides (PFPAs) can be covalently functionalized with graphene [75,76]. The functionalized graphene exhibits new chemical functionalities because the PFPAs groups impart solubility in both water and organic solvents [76].

Similarly, adsorption of aromatic molecules onto graphene, e.g., borazine (B_3_N_3_H_6_), triazine (C_3_N_3_H_3_), and benzene (C_6_H_6_) occurs through non-covalent bonds [77]. Park et al. [78] studied the influence of pyridine adsorption and the applied electric field on the band structure and metallicity of zigzag graphene nanoribbons (ZGNRs) using density functional theory. They found that adsorption of an electron-accepting organic molecule, such as pyridine, on ZGNRs should provide a simple and useful way to widen the band gap and can be used to turn the band structure of nanoscale electronic devices based on graphene applications.

Zhang et al. [79] developed a biosensor for the detection of microRNAs (miRNAs) based on graphene quantum dots (GQDs) and pyrene-functionalized molecular beacon probes (py-MBs). The pyrene unit served to shorten the distance between py-MBs and GQDs and to generate an increased fluorescence signal from dyes appended on the probes. When hybridized with the target miRNAs, the hairpin structure of py-MBs opened and formed more precise duplex structures.

Another important application of functionalized graphene is antimicrobial applications. Silver nanoparticles (AgNPs) be able to be decorated on the GO to make GO/Ag nanocomposite (Figure 4) [49,66]. This nanocomposite can be applied for coating and antimicrobial applications [49]. The ratio of D and G bands (I_D_/I_G_) of the GO/Ag nanocomposite may be elevated as a result of the disorder of the GO/Ag matrix [49,80].

Furthermore, Jeyaseelan et al. [81] developed the AGO for fluoride removal application, which was studied in terms of adsorption isotherms, kinetics (particle/intraparticle diffusion and pseudo-first/second-order models), and thermodynamic studies of AGO. The fluoride removal mechanism of AGO was found to be an electrostatic attraction.

Tissue engineering has emerged as an important approach to bone regeneration/substitution [82]. Functionalized graphene and its derivates have been also used in bone regeneration and tissue engineering. Graphene can be combined with natural and synthetic biomaterials to enhance the osteogenic potential and mechanical properties of tissue engineering scaffolds [83,84,85]. Scaffolds play a central role in tissue engineering as structural support for specific cells and provide the templates to guide new tissue growth and construction [84]. Nishida et al. [86] coated collagen scaffolds with various concentrations of GO and evaluated the bioactivity, cell proliferation, and differentiation both in vivo and in vitro. The results showed that GO affected both cell proliferation and differentiation and improves the properties of collagen scaffolds. Subcutaneous implant tests showed that low concentrations of GO scaffold enhance cell in-growth and are highly biodegradable, whereas high concentrations of GO coating resulted in adverse biological effects. Consequently, scaffolds modified with a suitable concentration of GO are useful as a bioactive material for tissue engineering.

Similarly, Kang et al. [87] studied the covalent conjugation of GO flakes to 3D collagen scaffolds improves the mechanical properties of the scaffolds and promotes the osteogenic differentiation of human MSCs (hMSCs) cultured on the scaffolds. The covalent conjugation of GO flakes to collagen scaffolds increased the scaffold stiffness by 3-fold and did not cause cytotoxicity. hMSCs cultured on the GO/collagen scaffolds showed significantly enhanced osteogenic differentiation compared to cells cultured on non-modified collagen scaffolds. The enhanced osteogenic differentiation observed on the stiffer scaffolds was mediated by MSC mechanosensing because molecules that are involved in cell adhesion to stiff substrates were either up-regulated or activated. The 3D GO/collagen scaffolds could offer a powerful platform for stem cell research and orthopedic regenerative medicine.

Recently, graphene-based bioactive glass is studied as a potential drug/growth factor carrier, which includes the composition–structure–drug delivery relationship and the functional effect on the tissue-stimulation properties [82,88,89]. Wang et al. [88] designed a scaffold composed of mesoporous bioactive glasses (MBG) and GO and studied the composite porous scaffold that promotes local angiogenesis and bone healing. This in vitro study found that the MBG/GO scaffolds have better cytocompatibility and higher osteogenesis differentiation ability with rat bone marrow mesenchymal stem cells (rBMSCs) than the purely MBG scaffold. Moreover, MBG/GO scaffolds promote vascular ingrowth and, importantly, enhance bone repair at the defect site in a rat cranial defect model. The new bone was fully integrated not only with the periphery but also with the center of the scaffold. Hence, the MBG/GO scaffolds possess excellent osteogenic-angiogenic properties which will make them appealing candidates for repairing bone defects.

Finally, biodegradable composites have been used in various regeneration processes applications such as the regeneration of bones, cartilage, and soft tissues. Stepanova et al. [90] synthesized aminated graphene with oligomers of glutamic acid and their use for the preparation of composite materials based on poly(ε-caprolactone) for tissue regeneration applications. The poly(ε-caprolactone) films filled with modified aminated graphene were produced and characterized for their mechanical and biological properties. They found that grafting of glutamic acid oligomers from the surface of aminated graphene improved the distribution of the filler in the polymer matrix that, in turn, improved the mechanical properties of composite materials. In addition, the modification improved the biocompatibility of human MG-63 osteoblast-like cells.

## 6. Graphene Coating Applications

The potential application of graphene for various biomedical applications is promising [3,9], such as anticorrosion, antibacterial coatings, and friction reduction [67], as shown in Figure 5. Graphene is chemically inert, atomically smoothness and high durability make it an alternative candidate for implant coatings [91].

### 6.1. Anticorrosion Coating

There are various applications of metallic materials in medicine and dentistry, such as dental implants, orthopedic fixations, orthodontic, joint replacements, stents, endodontic files, and reamers [92]. However, the disadvantage of such biomaterials is the metal ions release, such as Ni, Ti, Ag, hence, coating of metallic materials plays an important role in such problems [92,93]. Although various coatings are being tried on metallic biomaterials, especially nitinol (NiTi), producing a successful coating has been always a challenge [94,95,96,97,98,99,100,101,102,103,104,105,106,107]. Notable disadvantages of polymer coating include toxicity of the component’s roughness, porosity, and detachment of the coatings [108].

Even though graphene is an atom thick, it is inert and it is water-resistant and oxygen [4]. Hence, these properties combined with their durability and atomically stability has proven graphene to be useful as a corrosion barrier film [68,109,110,111,112]. Graphene can be directly grown on metallic surfaces (Mg, Zn, Ni, Al, etc.) to produce a protective coating [109,111,113]. Singh et al. [114] successfully developed an anti-corrosion graphene composite coating on Cu. In dentistry, graphene coatings can prevent corrosion of various metallic biomaterials such as archwires, files and reamers, and various metallic prostheses [65,68,109]. Furthermore, Hikku et al. [115] studied the anti-corrosion property of graphene and polyvinyl nanocomposite (GPVA) coating on the aluminum-2219 alloy (Al-2219). The corrosion rate for the coated Al-2219 alloys was better (polyvinyl alcohol coated alloy: 2.57 mm/year and GPVA coated alloy: 3.85 × 10^−4^ mm/year), whereas for untreated alloy: 45.25 mm/year in 3.5% NaCl solution (Figure 6). Hence, the GPVA coated Al-2219 alloy showed the best corrosion resistance than the uncoated alloy.

Graphene coatings can improve implant surface properties and reduce corrosion [91,116,117]. Podila et al. [91] produced graphene on Cu using chemical vapor deposition technique and transported it onto NiTi implant samples and studied the effects of the coatings on cell morphology and adhesion and they noted that the biological responses (cell adhesion and protein adsorption) were increased on the graphene-coated NiTi substrates, in comparison to the uncoated NiTi substrates. Thus, graphene-coated NiTi can be applied to the stent. Additionally, graphene could improve the osseointegration of Ti implants. In addition, Suo et al. [116] produced a homogeneous GO/chitosan/hydroxyapatite (GO/CS/HA) coating using electrophoretic deposition (EPD) on Ti. The GO/CS/HA coating’s wettability and bonding strength were greater than the HA, GO/HA, and CS/HA coatings. Moreover, the GO/CS/HA coating significantly enhanced the cell–material interactions in vitro and osseointegration in vivo. Hence, the GO/CS/HA coatings on Ti can be a potential coating in implant dentistry.

Magnesium (Mg) can be used to make biodegradable implants; however, its major drawbacks of difficult-to-control corrosion. Catt et al. [110] produced a conducting polymer 3,4-ethylene dioxythiophene (PEDOT) and a GO coating for Mg implants to prevent corrosion. It was found that the significant reduction of Mg ions concentrations and pH of the media from the PEDOT/GO coating suggests a significant corrosion resistance. A positive finding was that of decreased hydrogen amounts. Three important factors were due to the passive layer preventing the ingress of a solution, film’s negative charges, and development of a corrosion-resistant Mg-phosphate coat. Additionally, promising biocompatibility, in vitro, was observed as the coating did not show signs of toxicity to cultured neurons. Hence, the PEDOT/GO coating is successful in preventing Mg-based implants corrosion.

GO coating is also useful in tissue engineering and regenerative applications. Root fracture treatment, cementation of prostheses, pulp therapy, filling, repair, and regeneration of bone defects, may all indicate the use of bioactive cement. A bioactive cement typically releases calcium-ions (Ca^2+^), increases the alkalinity in its surrounding environment, and induces cell differentiation and formation of mineralized tissue. However, the cement tends to possess poor mechanical properties, at risk of fracture due to poor strength and fracture toughness [118]. The mechanical properties are improved by the addition of graphene. A doubling of the strength of 58S bioactive glass was observed by the addition of 0.5 wt.% [119]. The addition of GO [119] and rGO [120] have also shown significant improvements in mechanical parameters, the latter (rGO 1 wt.%) resulted in a 200% increase in the fracture toughness of hydroxyapatite [119]. Additionally, the bone cement’s bioactive properties are enhanced due to the addition of graphene. Several cell types, including bone marrow stem cells, periodontal ligament stem cells (PDLSCs), and dental pulp stem cells have shown spontaneous osteogenic differentiation as promoted by chemical vapor deposition-produced pristine graphene scaffolds and substrates [121,122]. Indeed, in vivo bone formation was exhibited by implanting GO-coated collagen scaffolds into tooth extraction sockets of beagle dogs. The GO-coated scaffolds showed increased bone formation and calcium absorption after 14 days, whereas the control scaffold was mostly filled with connective tissue [123]. Similarly, Zhou et al. [112] evaluated the bioactive effects of GO coated Ti substrate on PDLSCs and compared them to sodium titanate substrate. It was seen that the GO coated Ti substrate-induced PDLSCs exhibit suggestively higher alkaline phosphatase (ALP) activity, proliferation rate, and higher gene expression of osteogenesis markers, ALP, runt-related transcription factor 2 (Runx2), bone sialoprotein, and osteocalcin (OCN) compared to the sodium titanate substrate. Protein expressions of Runx2, bone sialoprotein, and OCN were additionally promoted by GO. Together, the findings suggest that GO and PDLSCs represent a favorable combination for regenerative medicine and dentistry.

### 6.2. Antibacterial Application

Bacteria and fungiform biofilms on the teeth surface, prostheses, or implant-anchored restorations [124]. If left untreated, the biofilm on dental implants may result in loss of the implant. It is challenging to produce implants with a high degree of osseointegration at the same time as inhibiting bacterial colonization [125,126,127]. The peri-implant diseases around implant result in implants failure due to supporting bone loss around the implant [128,129,130,131,132].

Various antimicrobial nanomaterials include polymers, nanoparticles such as gold nanoparticles (AuNPs), AgNPs, nanodiamond, and graphene-based materials [133,134,135]. Even though AgNPs show promising antibacterial properties, clinical applications of AgNPs are frequently impeded by their tendency to aggregate and consequent loss of antibacterial activity [133,136]. Additionally, the cytotoxicity of AgNPs towards human cells has been observed [137]. The amount of AgNPs should be minimal to avoid complications. However, AgNPs can be decorated onto GO to produce GO/Ag nanocomposite for increased antimicrobial activity [49,55]. AuNPs are used more for microbial identification rather than antimicrobial applications [138,139].

The graphene-based materials have powerful antimicrobial properties and inhibit bacterial colonization [69,125,140,141]. Agarwalla et al. [140] studied the graphene coating on Ti and their interaction with a biofilm of *Pseudomonas aeruginosa*, *Enterococcus faecalis*, *Streptococcus mutants*, and *Candida albicans*. They observed that when repeated twice, it reduces the formation of biofilm due to the hydrophobicity of graphene. These all findings show that coating Ti with graphene is useful for biofilm prevention on implants.

Graphene coatings enhance the adhesion of cells and osteogenic differentiation. Gu et al. [142] studied the osteoinductive and antibacterial effects of graphene sheets modified Ti implants. Chemical vapor deposition growth of graphene sheets by thermal treatment at 160 °C for 2 h and transferring to Ti discs. It was found that the graphene coatings on Ti enhanced adhesion of cells, osteogenic differentiation, and exhibited antibacterial properties. Similarly, another study also found similar results, i.e., osteogenic differentiation of mesenchymal stem cells using graphene [143]. Hence, graphene is capable to enhance the surface properties of NiTi-based implants.

Similarly, functionalized GO can improve the antimicrobial property, as demonstrated by the GO/Ag nanocomposite (Figure 4) [49,55]. Graphene nanocomposite has excellent antibacterial action against *Escherichia coli* and *Staphylococcus aureus* [55]. Zhao et al. [43] fabricated gelatin-functionalized GO (Gogel) surface coatings on NiTi substrates. The Gogel’s biocompatibility and antimicrobial properties were investigated, and it exhibited the highest rate of mouse osteoblastic adhesion, proliferation, as well as differentiation of cells compared to GO coated NiTi. Moreover, they reported that *E. coli* was suppressed on the surfaces of Gogel and GO. Following incubation on Gogel and GO, the integrity of the *E. coli* cell membrane was compromised and showed a low live/dead ratio. Therefore, GO-based coatings have both a high degree of biocompatibility and antimicrobial activity.

Chen et al. [144] studied the interaction of GO with four phytopathogens (two bacteria and two fungi). The studied bacteria were *Xanthomonas campestris pv. undulosa* and *pseudomonas* and studied fungus were *Fusarium oxysporum* and *Fusarium graminearum* (Figure 7). It was found that GO killed nearly 90% of the bacteria and repressed 80% macroconidia germination along with partial cell swelling and lysis at 500 μg mL^−1^. They mentioned that GO sheets intertwined the bacterial and fungal spores resulting in the local perturbation of their cell membrane, decreasing the bacterial membrane potential, and resulting in the leakage of electrolytes of fungal spores causing cell lysis (Figure 8).

## 7. Conclusions

The available literature shows that graphene-based coatings can improve the bioactivity of biomaterials, provide microbial- and corrosion-protection of implants, both in vitro and in vivo. Peri-implant infections causing peri-implantitis are among the most common reasons for implant loss and may be prevented by the coating of antimicrobial graphene. These additive properties of graphene can be modified by methods of functionalization. Graphene exhibits high biocompatibility, corrosion prevention, and antimicrobial properties to prevent the colonization of bacteria. Graphene coatings enhance adhesion of cells, osteogenic differentiation, and exhibit antibacterial activity to parts of Ti unaffected by the thermal treatment. Graphene-based materials are promising and may hold the key for the next material-based revolution for antimicrobial and coatings applications in dental and medical technology. More research is urged before clinical utilization will be a widespread reality.

## Figures and Tables

**Figure 1 ijms-23-00499-f001:**
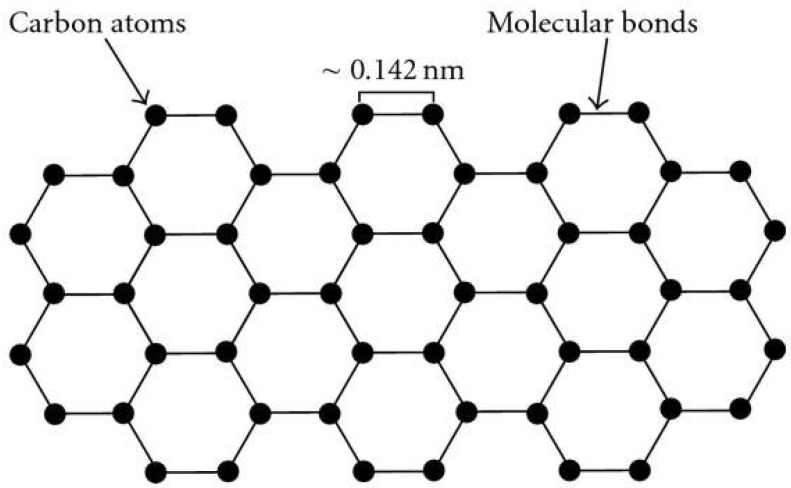
Structure of graphene [12].

**Figure 2 ijms-23-00499-f002:**
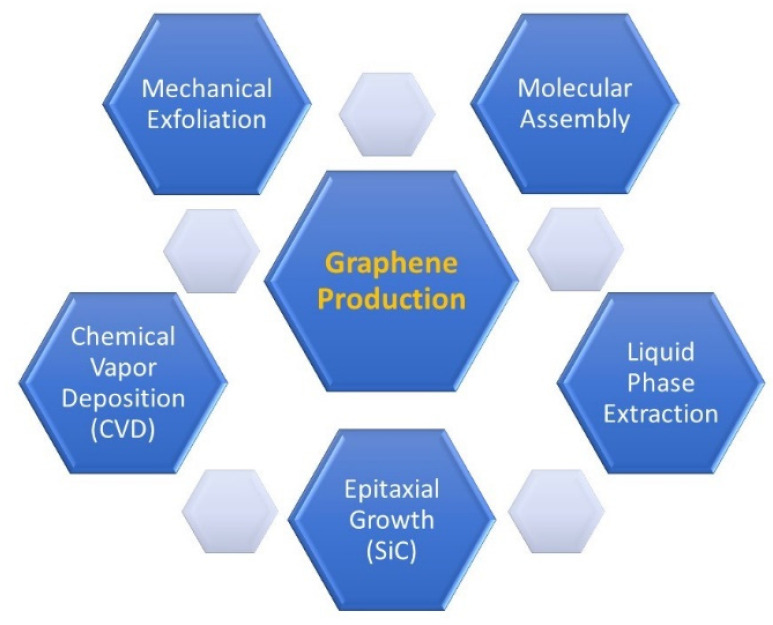
Various methods of production of graphene.

**Figure 3 ijms-23-00499-f003:**
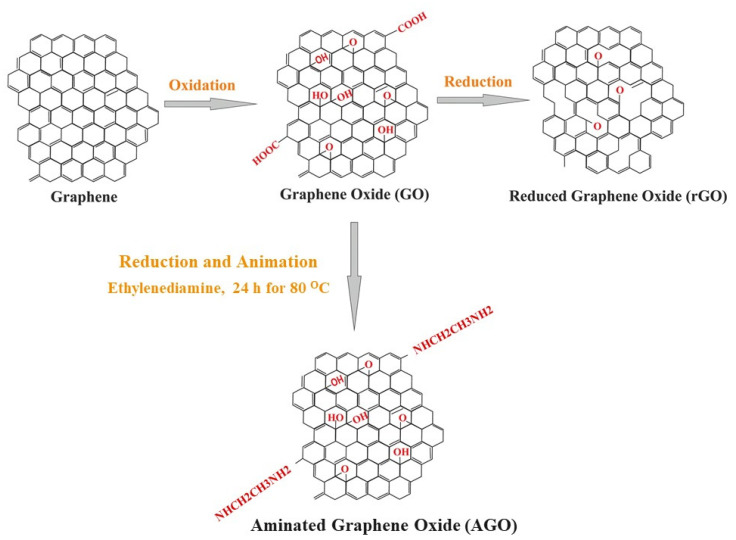
Production of graphene oxide (GO) reduced graphene oxide (rGO), and animated graphene oxide (AGO) from graphene [39].

**Figure 4 ijms-23-00499-f004:**
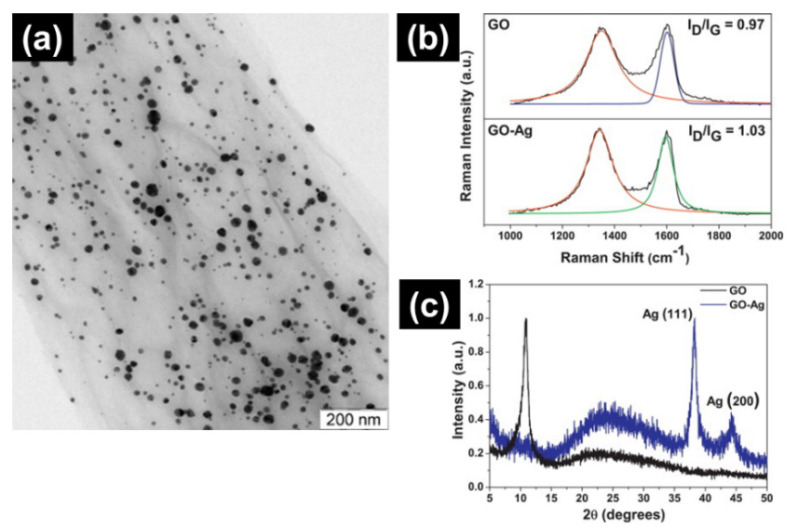
Characterization of graphene oxide (GO) nanocomposite formed from GO sheets decorated with Ag (GO/Ag). (**a**) Transmission electron microscopy (TEM) image, (**b**) Raman spectra, and (**c**) X-ray diffraction (XRD) [49].

**Figure 5 ijms-23-00499-f005:**
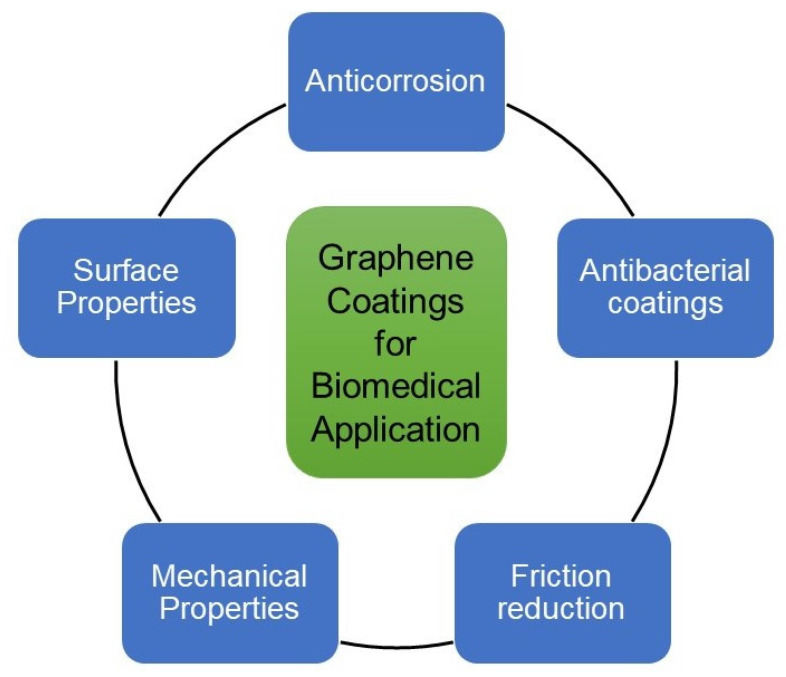
Biomedical applications of graphene-based coatings.

**Figure 6 ijms-23-00499-f006:**
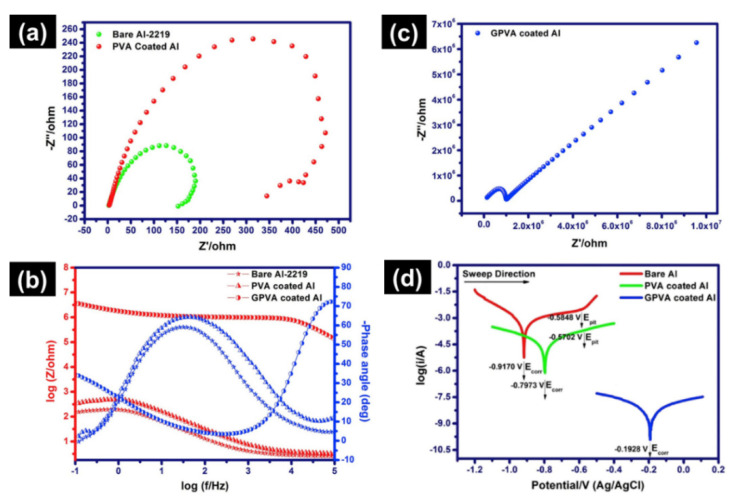
Anti-corrosion graphene blended with polyvinyl alcohol (GPVA) coatings: (**a**) Nyquist plot of bare Al-2219 and PVA coated Al-2219, (**b**) Nyquist plot of GPVA coated Al-2219, (**c**) Bode plot from the impedance analysis, and (**d**) Tafel plot of bare Al, PVA coated Al, and GPVA coated Al [115].

**Figure 7 ijms-23-00499-f007:**
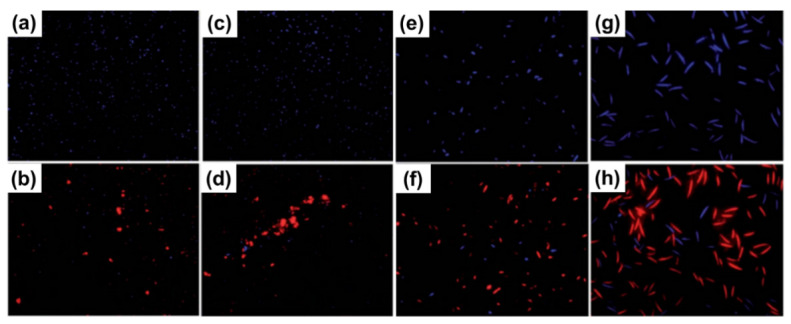
Fluorescence microscopy images of cells following exposure to graphene oxide (500 μg mL^−1^): (**a**) *X. campestris pv. undulosa*, (**c**) *P. syringae*, (**e**) *F. oxysporum*, and (**g**) *F. graminearum* and images following staining of cells with propidium iodide and fluorescence stain (**b**,**d**,**f**,**h**) [144].

**Figure 8 ijms-23-00499-f008:**
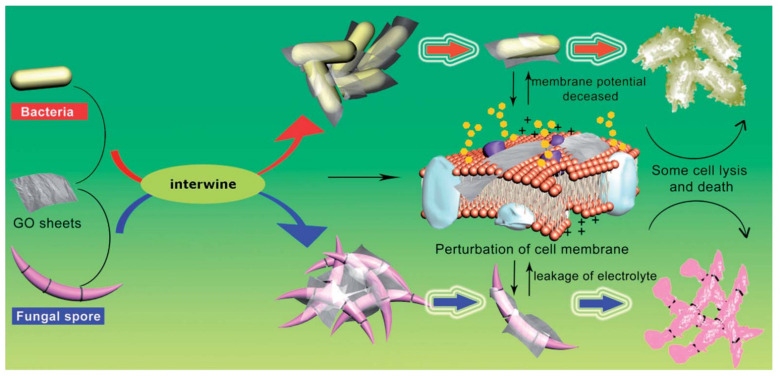
Antibacterial mechanism of graphene oxide against pathogens and fungal spores [144].

**Table 1 ijms-23-00499-t001:** Various methods of production of graphene and their properties [4].

Method	Crystallite Size (µm)	Sample Size (mm)	Charge Mobility (cm^2^ V^−1^ s^−1^)
CVD processed graphene	>1000	~1000	10,000
Mechanical exfoliation of graphene	>1000	>1	>2 × 10^5^ and 10^6^
Solution-processed graphene	~100	Infinite as a layer of graphene flakes	100
Epitaxial growth of graphene	50	100	10,000
Molecular assembly of graphene	<50	>1	NA

**Table 2 ijms-23-00499-t002:** Physical and mechanical properties of graphene [3,43].

Properties	Graphene	GO	rGO
Thermal conductivity	5000 W/m-K	2000 W/m-K	0.14–0.87 W/mK
Electrical conductivity	10^4^ S/cm	10^−1^ S/cm	200–35,000 S/cm
Electrical resistance	10^−6^ Ω-cm	NA	NA
Tensile strength	130 GPa	120 GPa	NA
Elastic modulus	1 TPa	0.22 TPa	NA
Poisson’s ratio	0.18	-	-

## Data Availability

Not applicable.

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
