# Peer review of "Graphene for Antimicrobial and Coating Application"

_ijms, 2022, doi:10.3390/ijms23010499_

Round 1

Reviewer 1 Report

In this contribution, Rokaya and collaborators reviewed antimicrobial applications of coatings derived from graphene. The topic is interesting and promising for future novel developments in the field of biomaterials. However, they are several flaws in the manuscript that need to be corrected before its publication in the International Journal of Molecular Science.

  1. The usage of English can be significantly improved. Several grammatical mistakes were detected throughout the entire manuscript (e.g., correct use of articles, conjugation of verbs in in singular/plural forms, etc.). English style could also be improved for a more fluid reading.

2. The manuscripts is in many parts very repetitive, which makes it somewhat difficult to read. For instance, the authors constantly repeat or stress the potential applications of the reviewed materials, but it is only until section 6 that they start discussing details about interesting reported applications of composites derived from graphene. Sections 1 to 5 show a sort of introduction into the topic, but it is very general and lacks of some specif details, but instead the authors constantly repeat the potential applications of these materials without providing details. In my opinion, the manuscript could be significantly optimized by avoiding mentioning so many times the discussed applications, but providing potential reading with more interesting details in the different sections from 1 to 5. For instance, they repeat several times the property of friction reduction of graphene based materials, but they do not provide any detailed discussion about this application, which I suppose it may not be so relevant in biomedial applications? Thus, they could avoid this kind of redundancies and go more straight to the main points.

3. The resolution of most of the Figures (i.e., plots and images) in the manuscript is very low; can the authors provide higher resolution files?

4. I would suggest the authors to cite the following contributions at the beginning of section 6.2 to provide potential readers with additional overviews about the state of the art of other antimicrobial nanomaterials and nanocomposites:

a) https://doi.org/10.3390/polym11111789

b) https://doi.org/10.1007/s10971-018-4890-9

Author Response

Response to Reviewer 1 Comments

In this contribution, Rokaya and collaborators reviewed antimicrobial applications of coatings derived from graphene. The topic is interesting and promising for future novel developments in the field of biomaterials. However, they are several flaws in the manuscript that need to be corrected before its publication in the International Journal of Molecular Science.

Thank you for your positive comments. Corrections in the Manuscript for Reviewer 2 are highlighted in Yellow color.

  1. The usage of English can be significantly improved. Several grammatical mistakes were detected throughout the entire manuscript (e.g., correct use of articles, conjugation of verbs in in singular/plural forms, etc.). English style could also be improved for a more fluid reading.

Response: Many thanks for the detailed feedback. We have now corrected grammatical mistakes and English style throughout the entire manuscript.

  1. The manuscripts is in many parts very repetitive, which makes it somewhat difficult to read. For instance, the authors constantly repeat or stress the potential applications of the reviewed materials, but it is only until section 6 that they start discussing details about interesting reported applications of composites derived from graphene. Sections 1 to 5 show a sort of introduction into the topic, but it is very general and lacks of some specif details, but instead the authors constantly repeat the potential applications of these materials without providing details. In my opinion, the manuscript could be significantly optimized by avoiding mentioning so many times the discussed applications, but providing potential reading with more interesting details in the different sections from 1 to 5. For instance, they repeat several times the property of friction reduction of graphene based materials, but they do not provide any detailed discussion about this application, which I suppose it may not be so relevant in biomedial applications? Thus, they could avoid this kind of redundancies and go more straight to the main points.

Response: The content of Section 1-5, is improved, and the redundancies are removed, focusing on main points.

  1. The resolution of most of the Figures (i.e., plots and images) in the manuscript is very low; can the authors provide higher resolution files?

Response: We have now improvised the figures as far as possible. Unfortunately, some of the figures that were adapted for the manuscript were not available in highest possible resolutions, resulting in lowering the images resolutions. 

  1. I would suggest the authors to cite the following contributions at the beginning of section 6.2 to provide potential readers with additional overviews about the state of the art of other antimicrobial nanomaterials and nanocomposites:
  2. a) https://doi.org/10.3390/polym11111789
  3. b) https://doi.org/10.1007/s10971-018-4890-9

Response: Many thanks for suggesting additional references, we have now added the recommended references. Please refer to section 6.2.

Reviewer 2 Report

Sapkota’s et al. manuscript is devoted to the overview of structure, properties and biological application of graphene and its derivatives. The manuscript is accurately prepared, well written and illustrated. Taking into account the large number of reviews available in the field of graphene and its applications, this review should be strengthened by adding relevant but missing information in the field of other graphene derivatives and their properties.

Major comments:

  1. The authors have discussed graphene, graphene oxide and reduced graphene oxide, but not a word was said about aminated graphene, which is also biocompatible. This information should be added and discussed.
  2. The information given in section 4 “Functionalization of graphene” is poor and does not provide the appropriate state-of-art in this field. It should be broadened with functionalization by polymers and small organic molecules for biological applications.
  3. Osteoinductive properties of graphene derivatives should be also discussed.

Author Response

Response to Reviewer 2 Comments

Thank you for your positive comments. Corrections in the Manuscript for Reviewer 2 are highlighted in Green color.

Sapkota’s et al. manuscript is devoted to the overview of structure, properties and biological application of graphene and its derivatives. The manuscript is accurately prepared, well written and illustrated. Taking into account the large number of reviews available in the field of graphene and its applications, this review should be strengthened by adding relevant but missing information in the field of other graphene derivatives and their properties. Major comments:

  1. The authors have discussed graphene, graphene oxide, and reduced graphene oxide, but not a word was said about it, which is also biocompatible. This information should be added and discussed.

Response: Details on aminated graphene is added and discusses.

  1. The information is given in section 4 “Functionalization of graphene” is poor and does not provide the appropriate state-of-art in this field. It should be broadened with functionalization by polymers and small organic molecules for biological applications.

Response: Functionalization of graphene Section is broadened with polymers and small organic molecules for biological applications.

  1. The osteoinductive properties of graphene derivatives should be also discussed.

Response: Osteoinductive properties of graphene derivatives are discussed.

Round 2

Reviewer 1 Report

I am satisfied with the implemented changes in the revised version of this contribution. Thus, I would recommend publishing this contribution in the International Journal of Molecular Science.

Reviewer 2 Report

Thank you very much for the revision. Now the manuscript looks much better and can be accepted for publication.